## PERSPECTIVE

# Excitatory neurotransmission at $O_2$ receptors in zebrafish gill: It takes two to tango

Colin A. Nurse[1] 
and Erin M. Leonard[2]

[1]*Department of Biology, McMaster University, Hamilton, ON, Canada*
[2]*Department of Biology, Wilfrid Laurier University, Waterloo, ON, Canada*

Email: nursec@mcmaster.ca

Handling Editors: Nathan Schoppa & Frank Powell

The peer review history is available in the Supporting Information section of this article (https://doi.org/10.1113/JP290062#support-information-section).

The ability to mount rapid cardiorespiratory responses to a decrease in $O_2$ availability (hypoxia) is critical for the survival of aerobic organisms. In adult mammals, these responses are initiated at specialized chemoreceptor (glomus) cells, located mainly in the richly-vascularized carotid bodies (CB), the primary arterial $O_2$ sensing organs. During acute hypoxia glomus cells depolarize, leading to $Ca^{2+}$-dependent release of excitatory (e.g. ATP, ACh) and inhibitory (e.g. dopamine) neurotransmitters that shape the afferent inputs into the brainstem (Leonard et al., 2018). However, in fetal/neonatal animals when CB chemoreceptors are immature, intrapulmonary neuroepithelial bodies (NEBs) are thought to act as the primary $O_2$ sensors (Cutz et al., 2013).

The evolution of these $O_2$ sensing pathways has generated considerable interest, beginning with the discovery >40 years ago of innervated 5-HT-containing neuroepithelial cells (NECs) in fish gill, a multifunctional organ that initiates cardiorespiratory reflexes in water-breathing vertebrates (Leonard et al., 2024). Significant progress has been made in elucidating the steps involved in hypoxia sensing by gill NECs; however, in contrast to the well-studied CB, the excitatory neurotransmitter pathways leading to hyperventilation and bradycardia in fish during hypoxia are poorly defined and even controversial (Leonard et al., 2024). Undoubtedly, progress has been hampered by the technical difficulty in recording afferent nerve activity emanating from the gill. In the current issue of *The Journal of Physiology*, Reed et al. (2025) circumvented these challenges using a novel transgenic zebrafish line, expressing the $Ca^{2+}$ reporter GCaMP in gill NECs, intrinsic sensory neurons (ChNs), and extrinsic sensory neurons in epibranchial and nodose ganglia (i.e. vagal sensory ganglia). This allowed *in vivo* (larvae) and *ex vivo* (whole gill) monitoring of hypoxia-induced $Ca^{2+}$ signals in these sensory elements.

The study by Reed et al. (2025) uncovered excitatory afferent pathways emanating from two NEC populations in the gill filament, one cholinergic (A-type) and the other serotonergic (S-type). The first step in the cholinergic pathway involved hypoxia-induced ACh release from A-type NECs acting on nicotinic AChR on the terminals of intrinsic ChNs. These ChNs functioned as interneurons, relaying afferent signals sequentially to neurons in the epibranchial and then the nodose ganglia. Consistent with this model: (1) exogenous ACh or nicotine mimicked the effects of hypoxia on ChNs and vagal neurons, and the presence of nicotine occluded the effects of hypoxia on ChNs; (2) hypoxia-induced $Ca^{2+}$ signals in ChNs were reversibly inhibited by the nicotinic AChR antagonist, hexamethonium; and (3) the $\alpha2$ nAChR subunit was localized by immunohistochemistry to nerve terminals adjacent to cholinergic NECs. By contrast, hypoxic stimulation of S-type NECs activated a parallel but separate pathway that bypassed the ChNs, resulting in direct excitation of vagal neurons. In these neurons, intracellular $Ca^{2+}$ signals were evoked by exogenous 5-HT and 5-HT3R agonists and the hypoxia-induced $Ca^{2+}$ signals were partially inhibited by the 5-HT3R blocker MDL72222.

Although these novel findings of two separate afferent pathways may reconcile some of the inconsistencies existing in the literature (Leonard et al., 2024), they raise additional questions about the physiological significance. As noted by Reed et al. (2025), functional S-type NECs appear to develop first in the distal tips of larval gill filaments, followed later by A-type NECs; accordingly, S-type NECs may function as the first line of defence against external hypoxic challenges when developing larvae rely mainly on diffusive gas exchange for $O_2$ uptake. This bears resemblance to the fetal mammalian lung where luminal serotonergic NEBs, which have a similar endodermal origin as S-type gill NECs, are well positioned to sense hypoxia in the fetal lung fluid by releasing 5-HT onto vagal nerve endings (Cutz et al., 2013). This supports the current view that mammalian NEBs (rather than CB chemoreceptors) are the true homologues of S-type gill NECs and probably provide the earliest defence of the fetus and newborn against hypoxia. Furthermore, Reed et al. (2025) propose that the later developing A-type NECs, also concentrated at the distal filament tips near the efferent filament artery, may function mainly as arterial $O_2$ sensors analogous to mammalian CB chemoreceptors. This model provides a plausible working hypothesis to explain the well-known ability of the fish gill to function as sensors of both environmental (water) $P_{O_2}$ and arterial (blood) $P_{O_2}$ (Leonard et al., 2024). In a recent study using the same zebrafish model hypoxia stimulated dopamine release from ChNs causing negative feedback inhibition of NECs (Reed & Jonz, 2025), now identified as the A-type subpopulation. This pathway shows a remarkable similarity to the mammalian CB where dopaminergic petrosal chemoafferents contribute to negative feedback inhibition of glomus cells during hypoxia (Leonard et al., 2018).

In conclusion, the elegant study by Reed et al. (2025) provides new insights into the excitatory pathways mediating the hypoxic ventilatory response in fish and our understanding of the evolution of $O_2$ sensing pathways. Gill NECs can sense other sensory chemostimuli such as $CO_2$, $NH_3$ and lactate (Leonard et al., 2024) and, although the two NEC populations may have overlapping functions, as in the study by Reed et al. (2025), it is presently unclear whether one shows a preference for one stimulus over another. In this regard, lactate is expected to act via the arterial circulation and thus may favor A-type NECs. Future studies, including a comparison of the $P_{O_2}$ *vs.* intracellular $Ca^{2+}$ dose–response curves could help clarify the physiological functions of S-type *vs.* A-type NECs. Also, the initial transduction steps in $O_2$ sensing in the two cell types may well be different given that specialized mitochondrial subunits are required for $O_2$ sensing by CB

The Journal of Physiology

arterial chemoreceptors compared to a membrane-associated NADPH oxidase for pulmonary NEBs (Cutz et al., 2013).

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

## Additional information

### Competing interests

No competing interests declared.

### Acknowledgements

EML acknowledges grant support from the Natural Sciences and Engineering Council of Canada (RGPIN-2023-05466).

### Keywords

ACh, fish gill, hypoxia, neurotransmission, neuroepithelial cells, 5-HT

## Supporting information

Additional supporting information can be found online in the Supporting Information section at the end of the HTML view of the article. Supporting information files available:

**Peer Review History**

