## [Peer Review History · The Journal of Physiology]

Excitatory neurotransmission at O₂ receptors in zebrafish gill: it takes two to tango

Colin A. Nurse and Erin Leonard
DOI: 10.1113/JP290062

Corresponding author(s): Colin Nurse (nursec@mcmaster.ca)

The following individual(s) involved in review of this submission have agreed to reveal their identity: Michael G Jonz (Referee #1)

Review Timeline:

Submission Date:

08-Sep-2025

Accepted:

18-Sep-2025

Senior Editor: Nathan Schoppa

Reviewing Editor: Frank Powell

Transaction Report:

Dear Professor Nurse,

Re: JP-P-2025-290062 "**Excitatory neurotransmission at O₂ receptors in zebrafish gill: it takes two to tango**" by Colin A. Nurse and Erin Leonard

We are pleased to tell you that your paper has been accepted for publication in The Journal of Physiology.

Please see below for a tiny amendment that can be made at proof stage.

Yours sincerely,

Nathan Schoppa
Senior Editor
The Journal of Physiology

If you would like to receive our 'Research Roundup', a monthly newsletter highlighting the cutting-edge research published in The Physiological Society's family of journals (The Journal of Physiology, Experimental Physiology, Physiological Reports, The Journal of Nutritional Physiology, and The Journal of Precision Medicine: Health and Disease), please click this link, fill in your name and email address and select 'Research Roundup':

<https://www.physoc.org/journals-and-media/membernews>

- You can help your research get the attention it deserves! Check out Wiley's free Promotion Guide for best-practice recommendations for promoting your work at: www.wileyauthors.com/eoo/guide. You can learn more about Wiley Editing Services which offers professional video, design, and writing services to create shareable video abstracts, infographics, conference posters, lay summaries, and research news stories for your research at: www.wileyauthors.com/eoo/promotion.

The Corresponding Author will receive an email from Wiley with details on how to register or log-in to Wiley Authors Services where you will be able to place an order

EDITOR COMMENTS

Reviewing Editor:

Thank you for this insightful commentary. There is only one spelling error noted by the authors that you need to fix.

Senior Editor:

Congratulations on acceptance of your Perspectives article! The reviewer, one of the authors of the manuscript, noted a minor spelling error that will need to be corrected by the authors in the Proofs stage.

REFEREE COMMENTS

Referee #1:

The authors provide an accurate and exciting summary of a recent study that will be published in the journal. They do an excellent job of contextualizing the meaning of the results, and their potential impact on our understanding of the evolution of oxygen sensing in vertebrates. The authors also end their article with a tantalizing suggestion, that A-type and S-type NECs might have different molecular sensors of oxygen. The perspectives article is a very interesting read. I have found only one minor item that needs correction below.

Line 59: Add an extra "2" to the chemical name. Should be "MDL72222".